

# A comparison of diceCT and histology for determination of nasal epithelial type

Timothy D. Smith[1], Hayley M. Corbin[2], Scot E. E. King[1],
Kunwar P. Bhatnagar[3] and Valerie B. DeLeon[4]

[1] School of Physical Therapy, Slippery Rock University, Slippery Rock, PA, USA
[2] Department of Biology, Slippery Rock University, Slippery Rock University, Slippery Rock, PA, United States
[3] Department of Anatomical Sciences and Neurobiology, University of Louisville, Louisville, KY, USA
[4] Department of Anthropology, University of Florida, Gainesville, Florida, United States

## ABSTRACT

Diffusible iodine-based contrast-enhanced computed tomography (diceCT) has emerged as a viable tool for discriminating soft tissues in serial CT slices, which can then be used for three-dimensional analysis. This technique has some potential to supplant histology as a tool for identification of body tissues. Here, we studied the head of an adult fruit bat (*Cynopterus sphinx*) and a late fetal vampire bat (*Desmodus rotundus*) using diceCT and μCT. Subsequently, we decalcified, serially sectioned and stained the same heads. The two CT volumes were rotated so that the sectional plane of the slice series closely matched that of histological sections, yielding the ideal opportunity to relate CT observations to corresponding histology. Olfactory epithelium is typically thicker, on average, than respiratory epithelium in both bats. Thus, one investigator (SK), blind to the histological sections, examined the diceCT slice series for both bats and annotated changes in thickness of epithelium on the first ethmoturbinal (ET I), the roof of the nasal fossa, and the nasal septum. A second trial was conducted with an added criterion: radioopacity of the lamina propria as an indicator of Bowman's glands. Then, a second investigator (TS) annotated images of matching histological sections based on microscopic observation of epithelial type, and transferred these annotations to matching CT slices. Measurements of slices annotated according to changes in epithelial thickness alone closely track measurements of slices based on histologically-informed annotations; matching histological sections confirm blind annotations were effective based on epithelial thickness alone, except for a patch of unusually thick non-OE, mistaken for OE in one of the specimens. When characteristics of the lamina propria were added in the second trial, the blind annotations excluded the thick non-OE. Moreover, in the fetal bat the use of evidence for Bowman's glands improved detection of olfactory mucosa, perhaps because the epithelium itself was thin enough at its margins to escape detection. We conclude that diceCT can by itself be highly effective in identifying distribution of OE, especially where observations are confirmed by histology from at least one specimen of the species. Our findings also establish that iodine staining, followed by stain removal, does not interfere with subsequent histological staining of the same specimen.

Corresponding author
Timothy D. Smith,
timothy.smith@sru.edu

## INTRODUCTION

Diffusible iodine-based contrast-enhanced computed tomography (diceCT) has emerged as a viable tool for discriminating soft tissues in serial CT slices, which can then be used for three-dimensional analysis. This has already been used to study multiple soft tissues, including muscle (*Cox & Jeffery, 2011*; *Gignac et al., 2016*; *Orsbon, Gidnark & Ross, 2018*; *Santana, 2018*; *Dickinson et al., 2019*), nervous (*Girard et al., 2016*), epithelial (*Gignac & Kley, 2014*; *Yohe, Hoffmann & Curtis, 2018*) and other tissues. The arrival of this technique holds much promise to replace the other traditional means of studying soft tissue structures, such as microdissection and histology, both of which are destructive techniques that permanently alter specimens (*DeLeon & Smith, 2014*; *Hedrick et al., 2018*). However, for many purposes even high resolution computed tomography currently lacks the ability to match histology in its capacity to identify extremely small anatomical structures (*e.g.*, *Reinholt et al., 2009*). In the present study we explore the capacity of diceCT for detecting internal nasal tissues. If diceCT can suffice for histology to some extent, the technique may have the major advantage to markedly decrease the laborious time involved in quantification or three-dimensional reconstructions using histology (*e.g.*, *Smith et al., 2007*; *Maier & Ruf, 2014*; *Yee et al., 2016*), while also providing increased sample sizes.

There are four commonly described types of epithelium that line the nasal cavity, of which two predominate (*Harkema, Carey & Pestka, 2006*; *Smith & Bhatnagar, 2019*). There are relatively small amounts of stratified epithelia that mainly line drainage routes and the vestibule, and a type of poorly known function called transitional epithelium. The vast majority of the nasal cavity is lined with respiratory and olfactory epithelia. In most mammals, respiratory epithelium is the predominant type anteriorly and inferiorly within the nasal cavities, and is recognizable based on pseudostratified, columnar structure, the presence of unicellular glands (goblet cells), and apical cilia that are observable by light microscopy (*Harkema, Carey & Pestka, 2006*; *Smith & Bhatnagar, 2019*). Olfactory epithelium (OE) is typically the predominant type posterodorsally; it is also pseudostratified, but has more numerous rows of nuclei throughout its thickness compared to respiratory epithelium. Most rows of nuclei are those of olfactory sensory neurons. Cilia are also present at the epithelial apex of OE, but they are enmeshed within a mucous covering that typically obscures them when viewed by light microscopy (*Dennis et al., 2015*; *Smith & Bhatnagar, 2019*). OE is generally thicker than non-olfactory types of epithelium (*Smith et al., 2021*). Using diceCT, Yohe and colleagues observed thickened epithelia along ethmoturbinals and other turbinals that bear most of the OE (*Yohe, Hoffmann & Curtis, 2018*). *Tahara & Larsson (2013)*, using diceCT to study quail visceral tissues, suggested both cellular density and cytoplasmic storage may promote radioopacity of epithelial tissues. Since OE is typically thicker than non-olfactory types (*Weiler & Farbman, 1997*), this suggests diceCT may be used in lieu of histology for identifying internal nasal tissues. However, *Yohe, Hoffmann & Curtis (2018)* also observed

that transition points between olfactory and respiratory epithelia are not detectable using diceCT alone. Nonetheless, these authors did observe some characteristics of the underlying lamina propria that helped to identify respiratory mucosa (specifically, glandular masses). This raises an important issue regarding olfactory tissues. In olfactory mucosa, there are glands present in the underlying connective tissue (or, lamina propria), intermingled with olfactory nerve bundles. Called Bowman's glands, these branched tubular masses are often densely packed (*Smith & Bhatnagar, 2019*). Based on basic characteristics of respiratory and olfactory mucosa, the glandular masses in the latter might be detected based on their uniform opacity, as contrasted to the more isolated "islands" of radiopaque masses that signify respiratory glands (*Yohe, Hoffmann & Curtis, 2018*).

Here, we studied the head of an adult fruit bat (*Cynopterus sphinx*) and a late fetal vampire bat (*Desmodus rotundus*) using high resolution diceCT. We seek to identify epithelial and mucosal (*i.e.*, epithelium + lamina propria) transition points using diceCT, with an added reference of histology subsequent to CT scanning. This method of examining individual specimens using both techniques was recently used to great advantage by *Girard et al. (2016)* to study murine brain lesions. In our methodological study, we assess the potential of diceCT to supplant histology as a tool for identification of olfactory mucosa.

## MATERIALS AND METHODS

Two bat species were selected, including two different ages. For a relatively large bat, an adult *Cynopterus sphinx* was included in our study. A far smaller specimen, a late fetal *Desmodus rotundus*, was also selected to determine limitations that may relate to size of the specimen. Both specimens are part of an archival collection of preserved and histologically sectioned bats in the collection of KPB, now curated by TDS. The *Desmodus* specimens in this collection, including a pregnant female with a late fetus, were originally collected in Veracruz, Mexico, in the 1980s (*Bhatnagar, 2008*). The *Cynopterus* specimen was collected in Jhabua, India (*Cooper and Bhatnagar, 1976*). Both specimens were originally fixed in 10% buffered formalin. The *Desmodus* fetus was fixed still within its amnionic sac (its mother was dissected open through the abdominal wall and uterine wall to enhance fixation). The *Cynopterus* was a full head and partial cervical region. Subsequent to fixation, both specimens were transferred to 80% ethanol and stored in the decades since, with periodic changes of fluid. Thus, the two specimens provided a parallel to common museum practice in terms of storage, and also two different stages of maturation and head size. Use of these specimens for the study was approved by the Institutional Animal Care and Use Committee at Slippery Rock University (IACUC protocol # 2021-03T).

Each specimen was scanned using traditional μ-CT and diceCT methods. Subsequently, each head was bisected and then serially sectioned in the coronal plane and stained with two procedures, hematoxylin-eosin and Gomori trichrome.

Conventional μ-CT scanning was conducted for the fetal *Desmodus* specimen at Northeast Ohio Medical University (NEOMED) using a Scanco vivaCT 75 scanner (scan

parameters: 70 kVp; 114 mA) and reconstructed with $0.0205 \times 0.0205 \times 0.0205$ mm cubic voxels. Conventional μ-CT scans of the adult *Cynopterus* specimen were collected at the University of Florida with a GE V|tome|xm 240 CT scanner (scan parameters: 100 kVp; 100 mA) and reconstructed with $0.0213 \times 0.0213 \times 0.0213$ mm cubic voxels. Specimens were prepared for diceCT at the University of Florida following the protocols outlined in *Gignac et al. (2016)*. Briefly, specimens were submerged in 20% sucrose solution for 24–48 h, and subsequently submerged in Lugol's iodine ($I_2KI$). The adult *Cynopterus* specimen was placed in a 5% Logol's solution for 7 days. The fetal *Desmodus* specimen was originally placed in a 5% Lugol's solution for 34 days, but this resulted in overstaining of the specimen and poor imaging outcomes. The specimen was destained by submerging in a 5% sodium thiosulfate ($Na_2S_2O_3$) solution. More recently we have optimized our diceCT protocol to include lower concentrations of Lugol's iodine (*e.g.*, 1%) over longer periods of time, refreshing the solution periodically. The fetal *Desmodus* specimen was again submerged in 20% sucrose solution for 48 h, and subsequently submerged in 1% Lugol's iodine for 7 days. DiceCT images were collected at the University of Florida GE V|tome|xm 240 CT scanner. The fetal *Desmodus* specimen was scanned using parameters of 160 kVp and 100 mA with a 0.5 mm copper filter and reconstructed with $0.0181 \times 0.0181 \times 0.0181$ mm cubic voxels. The adult *Cynopterus* specimen was scanned using parameters of 100 kVp and 140 mA (no filter) and reconstructed with $0.0256 \times 0.0256 \times 0.0256$ mm cubic voxels. Specimens were subsequently destained by submerging in a 5% sodium thiosulfate ($Na_2S_2O_3$) solution.

Histological sectioning was completed at the neurohistology laboratory in the School of Physical Therapy, Slippery Rock University. Each specimen was decalcified in a formic acid-sodium citrate solution with weekly tests to detect completion. After decalcification, the specimens were paraffin embedded, serially sectioned at 10 μm, and every 4th to 5th section was mounted and stained. All histological observations were conducted by TDS, using a Leica DMLB photomicroscope at X200 to X630.

A major step preceding cross-referencing histology and CT is the alignment of cross-sectional planes (Fig. 1). To do this, we identified corresponding features in the CT and histological data, and used these to reconstruct the plane of section in the CT volume. The CT volume was then digitally re-sliced using Amira 2019 software, such that orthogonal sections of the CT image corresponded to histological sections. In this study, all alignment was optimized for the caudal half of the nasal fossa, which contains most of the ethmoturbinal complex. A more detailed account of these methods is presented in *DeLeon & Smith (2014)*, and all alignments were conducted by VBD. Prior to observing diceCT slices, aligned diceCT slices were modified using the "auto contrast" function, *via* an automated batch command, using Adobe Photoshop software. This heightened the contrast of epithelium and lamina propria in the *Cynopterus* scan, but not appreciably so in the *Desmodus* scan.

In the first analysis, descriptive characteristics of nasal epithelia were assessed in the adult *Cynopterus* to add to existing data in the literature. All nasal tissues appeared exceptionally well-preserved. The observer (TS) identified olfactory mucosa according to the presence of rows of olfactory sensory neuronal bodies as is typical of OE, and the

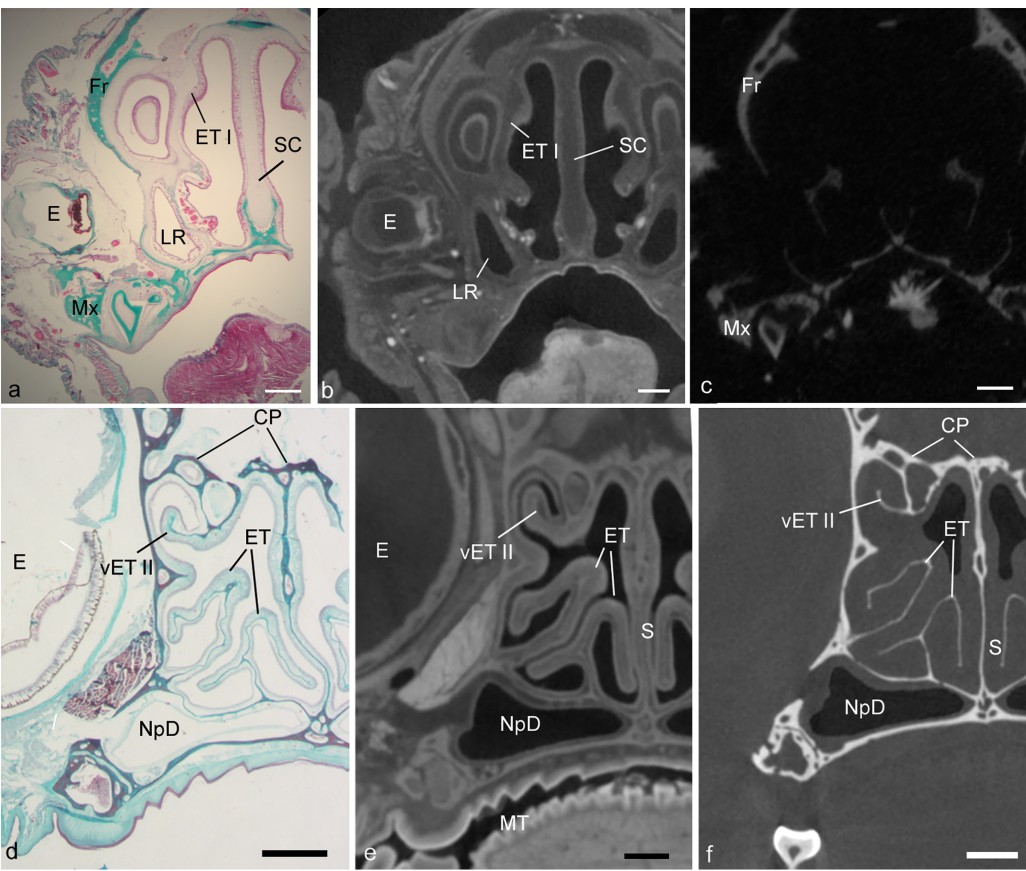

**Figure 1 Showing alignment of histology to CT.** Matching cross-sectional levels in ethmoturbinal region of the fetal *Desmodus* (A–C) and adult *Cynopterus* (D–F). Spatially dispersed structures viewed in a histological section of the fetal bat (A), such as the eye (E), first ethmoturbinal (ET I), septal cartilage (SC), maxilla (Mx), frontal bone (Fr), and or lateral recess (LR) can be seen in the diceCT scan (B) or μCT (C) slices. Note soft tissue structures of diceCT and osseous structures of μCT have been carefully aligned to histology. In the adult bat (D), histology reveals mucosa and supporting bones of the turbinals. Note the mucosal contours of ET I, and a ventral accessory lamella of ethmoturbinal II (vET II) seen in the diceCT slice (E) are in alignment with histology. Similarly, the bones of these turbinals seen in the μCT slices are well aligned with histology. Also, note the epT contour is clearly visible in the μCT slice, and thus was compared to diceCT and histology to assess shrinkage. CP, cribriform plates; NpD, naso-pharyngeal duct. Scale bars: a–c, 1 mm; d, 0.5 mm; e, f, 250 μm.

presence of Bowman's glands and olfactory nerves in the underlying lamina propria (*Harkema, Carey & Pestka, 2006*; *Smith & Bhatnagar, 2019*). Adjacent non-OE bore kinocilia, and was thus respiratory epithelium of varying morphology. Based on these characteristics, using the adult *Cynopterus*, thickness of OE was measured in ImageJ using X200 micrographs of seven sites (see Fig. S1), including: (a) dorsal rim of ethmoturbinal I, (b) lateral margin of nasal septum near its intersection with dorsal apex of the nasal chamber, (c) ventral rim of ethmoturbinal I, converging with ventral rim of ethmoturbinal II, (d) lateral margin of nasal septum, near its intersection with the palate, (e) medial margin of frontoturbinal 2, (f) dorsal rim of nasoturbinal, and (g) "roof" or dorsal apex of nasal fossa. Selected other sites were sampled for measuring to demonstrate the range of

thickness on non-OE, such as non-OE patches on the nasoturbinal, frontoturbinal, ethmoturbinal III, and a thick patch of non-OE found rostrally. For epithelial measurements, the sites in Fig. S1 and selected other sites were photographed in multiple sections (at every 16th to 32nd section, totaling 15 to 24 measurements per site) in which the structure was present. A photograph of a stage micrometer at the same ×200 magnification was used to set the scale in ImageJ. The height of the epithelium in each section was measured using a line tool drawn from the basal to apical sides of the epithelium, with the line oriented at a right angle to the basement membrane. A single factor analysis of variance (ANOVA) was used to assess whether significant ($p < 0.05$) differences exist in olfactory epithelial thickness among the five OE sites shown in Figs. S1A–S1G. In addition, we used t-tests comparing thickness (in μm) of olfactory epithelium (OE) *versus* non-olfactory epithelium (non-OE) on ethmoturbinal I (sites a *vs.* c) and the nasal septum (sites b *vs.* d). It should be noted that here the word "turbinal" is most frequently used to denote a mucosa-lined bony structure, rather than the bone itself, which would bear the same name.

The second analysis tested artifactual changes as a result of processing for diceCT and histology in the adult *Cynopterus* specimen. Multiple studies have observed artifactual changes to tissues with histological or diceCT methods. The dehydration steps that preceded paraffin embedding are known to produce extreme shrinkage artefacts manifested in stained sections (*Tahara & Larsson, 2013*; *DeLeon & Smith, 2014*). As noted above, shrinkage artifacts are also manifested following diceCT processing, but *Tahara & Larsson (2013)* assert diceCT-related shrinkage is similar to that resulting from fixation. Here, we expected shrinkage of tissues in both specimens due to the original treatment with a high concentration of Lugol's solution (*i.e.*, 5%). We sought to confirm this by measuring perimeters of selected structures visible in the μCT slices obtained using traditional scans, and then comparing these to the same measurements in matching diceCT slides, and to the histology sections to which both were aligned. Two sites were chosen for this analysis based on their isolation from other tissues: the maxilloturbinal and an epiturbinal (Figs. 2A, 2D). Paired t-tests were conducted to compare measurements of matching slice levels to assess for significant ($p < 0.05$) differences.

The third analysis tested the ability to discriminate OE and non-OE from diceCT images. The perimeter of the OE was measured on diceCT slices using ImageJ software. One observer (TS) annotated the histology-aligned diceCT slices by directly comparing them to matching histological sections. The histological section was viewed through on a monitor linked to a Axiocam MRc 5 Firewire camera attached to Leica DMLB compound microscope. Simultaneously, the matching diceCT slice was viewed on a second computer monitor using ImageJ software, and annotated according to the limits of olfactory epithelium on selected structures as determined by microscopic examination; the paint tool in ImageJ was used to annotate limits of olfactory epithelium based on matching contours (Fig. S2C). Two structures were selected for annotation of OE in both bat specimens: the combined septum/roof of the nasal fossa and the first ethmoturbinal (Fig. S2). These were annotated on each diceCT slice from the attachment site of the first ethmoturbinal as a caudal limit, and rostrally to the rostral limit of the olfactory mucosa.

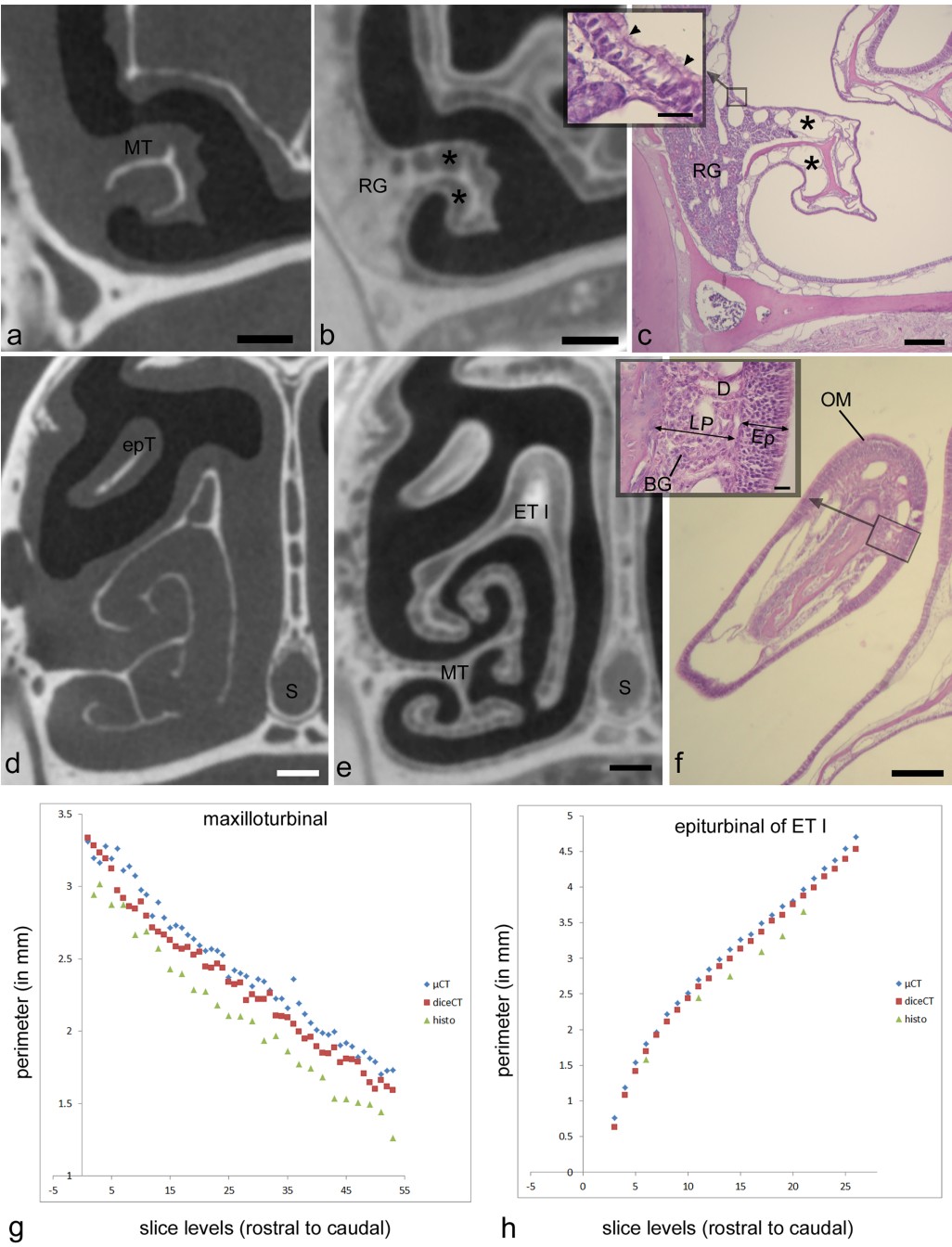

**Figure 2 Showing matching histology, diceCT, and microCT.** Paired measurements of identical structures in adult *Cynopterus* showing corresponding, aligned μ-CT slices (A, D), diceCT slices (B, E) and histological sections (C, F). (A–C) Cross-sections through a caudal portion of the maxilloturbinal (MT) at a matching level. Note the MT is more robust in the μ-CT slice (a) compared to the other images; the slightly thinner mucosa in diceCT (B) and histology (C) may correspond to shrinkage of the lamina propria, which contains mostly venous sinuses (*). (D–F) Cross-sections through the snout of *Cynopterus* revealing freely projecting epiturbinals (epT), an accessory projection of the first ethmoturbinal (ET I). A roughly similar contour is visible using all three imaging techniques. Note the respiratory epithelium is lined by a thin (scale = 20 μm) pseudostratified, ciliated epithelium (inset, c; arrowhead indicate cilia). Respiratory gland (RG) masses are seen near the root of the MT, and are isolated radioopaque masses in diceCT slices (B). The epT contains a dense lamina propria dorsally (F). A magnified view of the olfactory

**Figure 2 (continued)**
mucosa (OM) reveals the lamina propria (LP) is dense with Bowman's glands (BG), and the epithelium (Ep) is far thicker than that lining the MT (inset, f; scale = 20 μm). (G, H) Paired measurements of the MT and epT at matching levels, revealing that μ-CT measurements are larger compared to diceCT or (especially) histology at almost all matching levels. D, gland duct. Scale bars: a, b, d, e, 250 μm; c, 0.5 mm; f, 100 μm, insets, 20 μm.                                   

This region was selected because previous work on other mammals showed that transitions in epithelial type occur on both of these structures (*Smith et al., 2007*; *Pang et al., 2016*). Because histological measurements confirmed significant differences in OE *versus* non-OE thickness, as reported in other mammals (*Weiler & Farbman, 1997*), a second observer (SK) blindly annotated transitions from OE to non-OE based on changes in epithelial thickness in diceCT slices (Fig. S3), focusing on the region matching the histology series.

Subsequently, a second blind trial of the third analysis was performed which considered tissues deep to the epithelium. Mucous membranes, or mucosae, have two components that relate to its functional characteristics: the surface epithelium and the underlying, supportive lamina propria. Thus, a second trial was conducted blindly by SK, using an added criterion: characteristics of the lamina propria. *Yohe, Hoffmann & Curtis (2018)* observed that radiopaque glands may be indicative of respiratory mucosa. By viewing matched histological sections and CT slices, we observed this is also true of Bowman's glands. Thus, in a second trial, SK blindly annotated olfactory mucosa in diceCT slices based on the combined criteria of relatively thick epithelium and relatively higher radioopacity of the lamina propria deep to it. Because Kolmogorov–Smirnov tests revealed half of the data were not normally distributed, we transformed all the data ($\log_{10}$) prior to analysis. The measurements were compared to the blindly annotated perimeters in matching series of sections using a repeated measures two-way (ANOVA) testing the effects of location and annotation type in SPSS software. *Post hoc* testing for between-groups differences was done using Fisher's Least Significant Differences (LSD) test.

# RESULTS

## Alignment of CT and histology

Alignment of μCT and diceCT volumes to the plane of histology resulted in an excellent correspondence of structures throughout the head in the fetal *Desmodus*. Some shrinkage of mucosa in histology made the airways appear larger in cross-section (Fig. 1A), but contours matched well (Figs. 1A–1C). Alignment of μCT and diceCT volumes to the plane of histology in the adult *Cynopterus* resulted in excellent correspondence of structures in some regions, but rostrally the matching of contours was less precise. This was particularly so with respect to free projections of turbinals, which are known to shrink more than attached structural elements (*DeLeon & Smith, 2014*). However, very precise contour matching was accomplished in the more caudal olfactory region (Figs. 1D–1F). All CT scan slices are available on MorphoSource at the project link (https://www.morphosource.org/projects/000365326). The examination of diceCT and

**Table 1 Descriptive statistics of thickness (in μm) of olfactory epithelia by site and results of single factor ANOVA.**

| Groups | Count | Sum | Average | Variance | | |
|---|---|---|---|---|---|---|
| ETI | 24 | 1,011.75 | 42.156[1] | 29.505 | | |
| Nasal septum | 23 | 1,152.95 | 50.128 | 15.544 | | |
| FT | 9 | 453.75 | 50.417 | 31.676 | | |
| NT | 15 | 664.13 | 44.275[1] | 55.098 | | |
| "roof" | 14 | 813.67 | 58.119 | 32.733 | | |
| **ANOVA** | | | | | | |
| *Source of Variation* | *SS* | *df* | *MS* | *F* | *P-value* | *F crit* |
| Between groups | 2,614.917 | 4 | 653.7291 | 21.16572 | 6.22E−12 | 2.485885 |
| Within groups | 2,470.898 | 80 | 30.88622 | | | |
| Total | 5,085.814 | 84 | | | | |

Note:
[1] LSD comparisons reveal epithelia of ET I and NT are significantly thinner compared to all other olfactory sites, but not significantly different from each other.

histology in matching planes, from the same specimens, provided an ideal opportunity to confirm tissue identity based in microanatomical characteristics (see below).

## Analysis 1: epithelial metrics in adult *Cynopterus* based on histology

In the adult *Cynopterus*, most non-OE of the nasal cavity (excluding the vestibule) is ciliated columnar or ciliated pseudostratified columnar in morphology, with a broad range of thickness, from 6.3 to 51.7 μm. However, the thickest patch of non-OE was restricted to a zone just rostral to the first ethmoturbinal, and also rostral to the most rostral appearance of olfactory mucosa (as verified using histology). Aside from this patch, the thickest non-OE was 21.2 μm. Olfactory mucosa ranged from 26.3 to 71.6 μm in thickness based on a sampling of multiple turbinals and other surfaces. Among five locations of OE measured, a one-way single-factor ANOVA reveals significant ($p < 0.001$) differences based on site of measurement (Table 1). More specifically, LSD tests reveal epithelia of ET I and nasoturbinal are significantly thinner compared to all other olfactory sites, but not significantly different from each other (Table 1). One apparent trend in *Cynopterus* is that the free margins of turbinal projections have relatively thinner OE, whereas measurements taken from the septum, the roof of the nasal fossa, and along planar surfaces of turbinals are thicker.

To assess whether thickness of epithelial types (olfactory and non-olfactory) on individual structures is distinctive, we compared selected histological measurement sites on the first ethmoturbinal and on the nasal septum (sites "a" *versus* "c" and "b" *versus* "d" from Fig. S1). Independent t-tests, assuming unequal variance, indicate OE is significantly thicker on both structures (Table 2).

## Analysis 2: assessment of epithelial perimeter, and artifactual changes following iodine and histological processing

Two structures that were distinctly visible in μ-CT scan slices (Figs. 2A, 2D; maxilloturbinal and epiturbinal) were measured and then remeasured in matching, aligned

**Table 2 Results of t-tests comparing thickness (in μm) of olfactory epithelium (OE) *versus* non-olfactory epithelium (non-OE) on ethmoturbinal I and the nasal septum.**

| Comparison | | Mean | Variance | t statistic | p value |
|---|---|---|---|---|---|
| ethmoturbinal I: | OE | 42.16 | 29.51 | 24.85 | *p* < 0.001 |
| | non-OE | 10.89 | 7.43 | | |
| nasal septum: | OE | 50.13 | 16.08 | 32.41 | *p* < 0.001 |
| | non-OE | 16.08 | 8.99 | | |

diceCT slices (Figs. 2B, 2E) and histological sections (Figs. 2C, 2F) of the adult *Cynopterus*. We expected that tissue shrinkage due to processing would yield perimeters that are greater in measurements taken from μ-CT slices compared to diceCT or histology.

μ-CT–based measurements of the maxilloturbinal are an average of 0.13 mm greater (~5% difference) in perimeter compared to the same measurement in diceCT slices (Table S1). μ-CT-based measurements of the epiturbinal, an accessory flange of ET I, are an average of 0.11 mm greater (~3% difference) in perimeter compared to the same measurement in diceCT slices (Table S2). A paired t-test reveals that measurements of the maxilloturbinal from μ-CT slices are significantly greater than those from matching diceCT slices (t = 11.1; *p* < 0.001). Similarly, a paired t-test reveals that measurements of the epiturbinal from μ-CT slices are significantly greater than those from matching diceCT slices (t = 18.59, *p* < 0.0001). When matching μ-CT and diceCT slice levels plotted against one another, measurements are nearly parallel (Figs. 2G, 2H), and μ-CT slice measurements are mostly but not always greater than those from diceCT. The parallel nature of measurements, when plotted against matching slice levels, suggests the slices are well-aligned and that the difference is consistent.

The number of paired comparisons of measurements from histology relative to matching μ-CT slices were fewer, since not all sections were used for staining. However, comparisons of data from matching sections suggest an even greater disparity between measurements based on μ-CT *versus* histology at matching levels. μ-CT–based-measurements of the maxilloturbinal are an average of 0.33 mm greater (~14% difference) in perimeter compared to the same measurement in matching histological sections (Table S1). μ-CT–based measurements of the epiturbinal, an accessory flange of ET I, are an average of 0.33 mm greater (~11% difference) in perimeter compared to the same measurement in histology slices (Table S2). A paired t-test reveals that measurements of the maxilloturbinal from μ-CT slices are significantly greater than those from matching histology sections (t = 22.27, *p* < 0.0001). A paired t-test reveals that measurements of the epiturbinal from μ-CT slices are also significantly greater than those from matching diceCT slices (t = 13.24, *p* < 0.0001).

### Criteria for identifying olfactory mucosa using diceCT

Rough qualitative comparisons of epithelial thickness are possible using diceCT, and sometimes reveal the approximate limits of OE (*e.g.*, Figs. S2C, S2D). However, since epithelial thickness sometimes falls close to the CT voxel dimensions, at least based on histology (Tables 1, 2), we expected that the diceCT images in this study might lack the

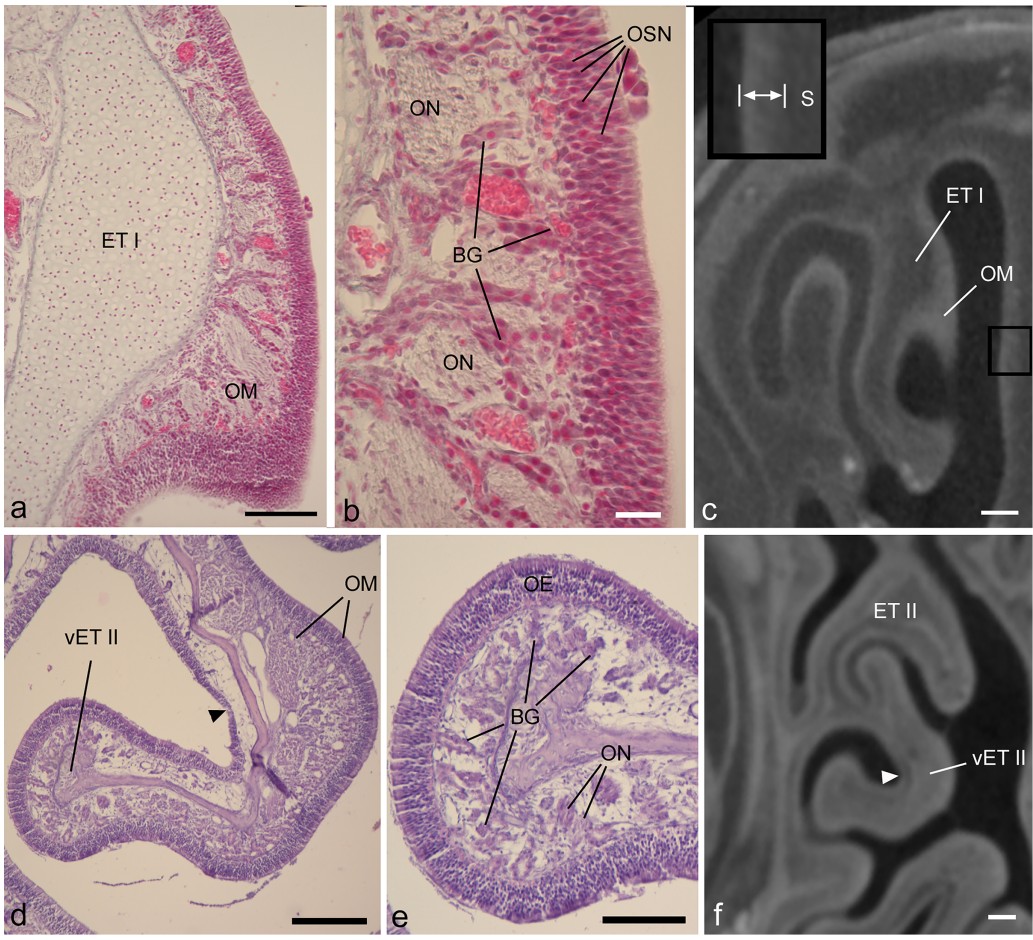

**Figure 3 Showing diceCT aspects indicating mucosal identity.** Matching histology and diceCT of *Desmodus rotundus* (A–C) and *Cynopterus sphinx* (D–F), revealing characteristics of olfactory mucosa that are apparent in diceCT. (A) The first ethmoturbinal (ET I) shown in a Gomori-trichrome stained section at its point of attachment to the nasal fossa "roof." On its medial face is a thick olfactory mucosa (OM). (B) An enlarged view of the OM. OM thickness corresponds in part to an olfactory epithelium in which the bodies of sensory neurons are staggered throughout its depth; note rows of nuclei (OSN). A greater extent of its thickness corresponds to the lamina propria, which is home to numerous Bowman's glands (BG) and olfactory nerves (ON). (C) The same turbinal shown in an aligned diceCT slice. The entire mucosal depth of ET I is radiopaque on its medial side. The septum that faces ET I has a thinner mucosa. This mucosa (enlarged in inset, space between bars) has a greater radioopacity than the septal cartilage (S) that supports it. (D) A ventral accessory lamella of ethmoturbinal II (vET II) lined with olfactory OM. (E) An enlargement of the free margin of this turbinal revealing a thick olfactory epithelium (OE) and densely glandular lamina propria. (F) An aligned diceCT slice of ET II, showing the turbinal is almost completely radioopaque. Note, however, small patches of non-OE are easily identifiable using histology (d, black arrowhead), but are less distinct in diceCT (f, white arrowhead). Also note, in many locations the thickness of the olfactory epithelium in the fetus closely approaches the voxel dimension of ~18 μm (B). In contrast, the olfactory epithelium in *Cynopterus* more greatly exceeds the voxel dimension of ~21 μm (E). Scale bars: a, 100 μm; b, 20 μm; c, 250 μm; d, 200 μm; e, 100 μm; f, 0.5 mm.

resolution to establish epithelium type based on epithelial thickness alone. Nonetheless, our diceCT-histology matches indicate that OE may be qualitatively identified by its thickness and high degree of radioopacity compared to non-OE (Figs. 2E, 3C).

The degree of radioopacity may relate to density of nuclei of sensory neurons (Fig. 3). Thus, thickness and radioopacity were two criteria used in our attempt to identify the boundaries between non-olfactory and olfactory epithelia.

Most olfactory portions of ethmoturbinals have relatively thick OE and underlying connective tissue (lamina propria). In both the fetal and adult bat, thickness of olfactory mucosa (epithelium and lamina propria) is greatest on the medial side of ethmoturbinals (Figs. 3A, 3D). These parts of the turbinals have epithelia that exhibit staggered locations of olfactory sensory neurons throughout epithelial depth, as evidenced by the rows of nuclei (Fig. 3B), and the lamina propria is packed with Bowman's glands and olfactory nerves (Figs. 3B, 3E). A comparison to corresponding, aligned diceCT slices reveals that the olfactory mucosa lining these parts of the ethmoturbinals are highly radiopaque (Figs. 3C, 3F), especially by comparison to more ventrally positioned structures such as the maxilloturbinal (Figs. 2B, 2E). Such lamina propria can appear uniformly opaque in diceCT (*e.g.*, Fig. 3F) or may have a "mottled" appearance with radioopaque patches just deep to mucosal surface; these patches are visible even where the epithelium is indistinct (Fig. 3F, inset).

Just as thickness of olfactory epithelium varies (Tables 1, 2), so does thickness of the lamina propria deep to it. For example, histology confirms convex sides of turbinals have a thicker, more densely glandular lamina propria than concave (meatal) surfaces (Fig. 3D). Nevertheless, in matching diceCT slices, the lamina propria is radioopaque on both sides (Fig. 3F). Some small patches of non-OE that interrupt the continuity of OE may be difficult to detect. While easily identifiable using histology (Fig. 3D, black arrow), they are less distinct in diceCT (Fig. 3F, white arrow).

## Analysis 3: perimeter of olfactory surfaces in the region of the rostral part of ethmoturbinal I

### Trial 1: assessing thickness changes in epithelial thickness using diceCT

Blind annotations of epithelial changes from OE to non-OE in diceCT (by coauthor SK, blind to histology) were mostly successful in the case of the adult *Cynopterus*. For the most part, measurements of OE perimeter in slices annotated according to changes in epithelial thickness alone closely track measurements of slices based on histologically-informed annotations (coauthor TS, based on histology) (Fig. S3). However, for both the ethmoturbinal and the septum/roof, "blind" annotations of diceCT images overestimated the amount of OE at the rostral end. This suggests thicker non-OE exists rostrally, which was verified by examination of histology in this region (Fig. S2I). Thus, the majority of data points are parallel between diceCT and histological annotated series, but rostrally the perimeters diverge (see right side of plot in Fig. S2E).

In the adult *Cynopterus*, repeated measures two-way ANOVA reveals significant differences between perimeters measured based on blind annotations *versus* histology-informed annotations of diceCT slices based on location (F = 9.193; $p < 0.01$), but not annotation type (F = 1.205; $p > 0.05$), or interaction (F = 1.27; $p > 0.05$). Perimeters of OE in the roof/septum measured by the two methods differ by 0.52 mm on average, with slices annotated blindly measuring less. This difference accounts for 26% of the

average OE perimeter measured from histology-annotated slices. Blind annotations of the ethmoturbinal yield OE perimeter measurements that are 0.16 mm less, on average, than slices annotated according to histology (Table S3). This difference accounts for 10% of the average OE perimeter measured from histology-annotated slices.

In the fetal *Desmodus*, repeated measures two-way ANOVA reveals significant differences between perimeters measured based on blind annotations using diceCT *versus* histology of structures based on location (F = 12.667; $p < 0.01$), and based on type of annotation (F = 49.864; $p < 0.001$), but no significant interaction effect (F = 0.106; $p > 0.05$). Blind annotations of the roof/septum yielded OE perimeter measurements that were 0.62 mm less, on average, than diceCT slices annotated with reference to histology (Table S4). This difference accounts for 17% of the average OE perimeter measured from histology-annotated diceCT slices. Blind annotations of the ethmoturbinal yielded OE perimeter measurements that were 0.47 mm greater, on average, than diceCT slices annotated according to histology (Table S4). This accounts for 22% of the average OE perimeter measured from histology-annotated slices.

### Trial 2: assessing thickness changes in mucosa using diceCT

When the epithelium and lamina propria are considered together, side-by-side comparison of perimeters measured from diceCT slices compared to matching histological sections reveal a closer match. Most of the radiopaque, thick mucosa on diceCT corresponds to olfactory mucosa as verified using histology (Fig. 3). The perimeter of OE on the first ethmoturbinal and the nasal roof/septum was annotated in a second trial using the combined criteria of epithelial thickness and degree of radioopacity of the lamina propria.

In the adult *Cynopterus*, repeated measures two-way ANOVA reveals significant differences between perimeters measured based on blind annotations using diceCT *versus* histology of structures based on location (F = 30.4; $p < 0.01$), annotation type (F = 19.13; $p < 0.01$), and a significant interaction (F = 5.002; $p < 0.05$). In both the ethmoturbinal and the septum/roof, perimeters blindly annotated for olfactory mucosa limits closely parallel measurements informed by histology (Fig. 4). Blind annotations of the ethmoturbinal yield olfactory mucosa perimeter measurements that were 0.04 mm greater, on average, than slices annotated according to histology (Table S5). This accounts for 2% of the average olfactory mucosa perimeter measured from histology-annotated slices. Blind annotations of the roof/septum yield perimeter measurements that were 0.24 mm greater, on average, than slices annotated according to histology (Table S5). This difference accounts for 10% of the average perimeter measured from histology-annotated slices.

In the fetal *Desmodus*, repeated measures two-way ANOVA reveals significant differences between perimeters measured based on blind annotations using diceCT *versus* histology of structures based on location (F = 17.112; $p < 0.01$), but no significant difference based on type of annotation (F = 0.762; $p > 0.05$) or interaction effect (F = 0.119; $p > 0.05$). Blind annotations of the roof/septum yield olfactory mucosa perimeter measurements that were 0.06 mm greater, on average, than slices annotated according to

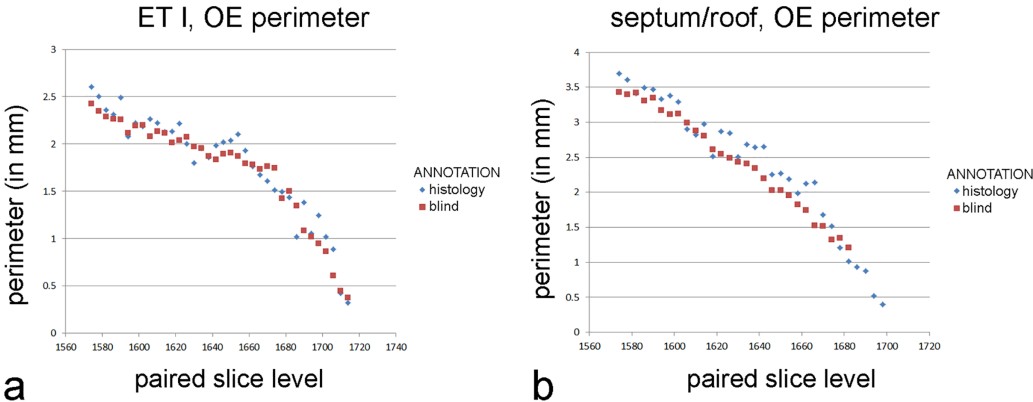

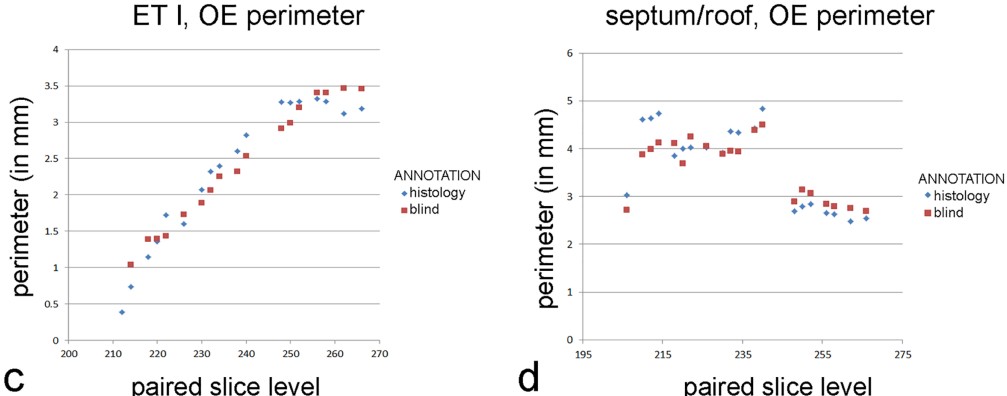

**Figure 4 Graphs of blind *versus* histology-informed annotations.** Perimeter measurements of olfactory epithelia in the two bats. (A, C) Ethmoturbinal I (ET I); (B, D) septum and roof. Measurements of matching levels are plotted for slices annotated according to histological observations (blue symbols) and those annotated blindly based on mucosal appearance in diceCT.

histology (Table S6). This difference accounts for 2% of the average perimeters measured from histology-annotated slices. Blind annotations of the first ethmoturbinal yield perimeter measurements that were 0.09 mm greater, on average, than slices annotated according to histology (Table S6). This accounts for 2% of the average OE perimeter measured from histology-annotated slices. Nevertheless, for both sites, comparing measurements by slice shows that "blind" annotations of olfactory mucosa closely parallel measurements informed by histology (Fig. 4).

## DISCUSSION

The increased availability of high resolution µCT has offered a valuable means of studying minute anatomical structures. Perhaps the greatest benefit has been the ability to non-destructively study rare specimens (*Hedrick et al., 2018*; *Yapuncich et al., 2019*); this is particularly of value for the study of endangered species or valuable museum samples (*Smith et al., 2020*). Recently, the availability of diceCT has provided an innovative means

to virtually dissect soft-tissue structures, such as individual muscles (*e.g.*, *Santana, 2018*; *Dickinson et al., 2019*, *2020*) or visceral structures (*e.g.*, *Vickerton, Jarvis & Jeffery, 2013*). Still, μCT has not achieved the resolution necessary to match histology in efforts to establish the finest osteological features, such as microscopic bony bridging across sutures at early points of fusion, while still maintaining overall spatial context (*Reinholt et al., 2009*). Similarly, although diceCT discriminates muscles and organs based on anatomical context, it does not allow discrimination of specific connective epithelial tissues (*Yohe, Hoffmann & Curtis, 2018*). For these aims, histology remains the gold standard.

However, in the same way that gross anatomical spatial relationships allow the identification of specific muscles in diceCT studies (*Santana, 2018*; *Dickinson et al., 2019*), microanatomical elements of body tissues also provide critical context to infer tissue types. Here, we show that the tissue composition of the lamina propria varies, and the thickness of the nasal epithelia varies, in such a way to make identification of olfactory mucosa possible using diceCT.

## Artefactual changes to specimens relating to diceCT and histological processing methods

DiceCT has the potential to resolve several drawbacks to the use of histology to study vertebrate microanatomy. Most notably, the iodine immersion involved in diceCT is reversible, at least to a great extent (*Girard et al., 2016*), whereas histology permanently limits the use of tissues for study using most other methods, aside from bright-field microscopy and three dimensional reconstruction; *i.e.*, it is a highly destructive technique that is not ideal for examination of rare specimens (*DeLeon & Smith, 2014*; *Hedrick et al., 2018*). Even though histological sections present unparalleled microscopic detail, the cross-sectional plane is permanent, whereas computed tomography data may be manipulated for multiplanar examination (*DeLeon & Smith, 2014*). Moreover, histological processing introduces artefactual changes, such as tissue shrinkage and folding (*Rolls & Farmer, 2008*). When used for three-dimensional reconstructions, this leads to quantifiable distortions (*e.g.*, reduced length dimensions) of structures (*DeLeon & Smith, 2014*). While this can be corrected (*Smith & Bhatnagar, 2019*), diceCT remains a potential alternative. However, diceCT also introduces artifacts such as tissue shrinkage (*Hedrick et al., 2018*), as demonstrated by the results here.

Before discussing the extent of shrinkage to which diceCT or histology may induce on nasal structures, we should acknowledge uncertainty regarding the extent of shrinkage caused by fixation and long-term ethanol storage. Multiple studies have documented that ethanol storage, especially long-term storage, causes marked shrinkage in soft tissue structures (*e.g.*, *Hedrick et al., 2018*; *Leonard et al., 2021*). Some structures appear more susceptible than others (*e.g.*, eyes), but it is also demonstrable that intact, whole animal specimens shrink less than isolated organs or tissue blocks (*Fox et al., 1985*). We might infer that undecalcified bone of whole specimens is the most important tissue that resists shrinkage, since bone as a tissue shrinks far less than soft tissue organs during histological processing that involves dehydration (*Buytaert et al., 2014*). And we also suspect that immature specimens with less fully calcified bones shrink more than adult

samples. With the uncertainty regarding the extent of shrinkage in mind, the samples used in the present study are very similar to museum samples in that they have been stored in ethanol for decades. In that respect, the results inform us as to the potential value of diceCT for studying museum fluid collections.

Here, we provide quantitative confirmation that both diceCT and histology result in tissue shrinkage of nasal tissues, as is known for other regions/structures (*e.g.*, *Hedrick et al., 2018*). We draw this inference based on a comparison to μ-CT slices, which can fortuitously allow examination of soft tissue contours within the nasal cavity. Recently, *Smith et al. (2021)* were able to examine mucosal surfaces in a cadaveric dog snout in high resolution μ-CT scan slices, and even measure mucosal thickness, demonstrating some utility of μ-CT for soft tissue studies, though epithelia were not observable. Several mucosal structures are visible in our samples. Most mucosal contours were obscured, likely due to the presence of fluid, which we assume may be more apt to remain in the snouts of small mammals. However, two structures (the maxilloturbinal and an epiturbinal) fortuitously had exposed contours, enabling a comparison of these structures across corresponding μ-CT, diceCT, and histology sections.

Shrinkage artifacts are a well-known artifactual change associated with the diceCT procedure (*Tahara & Larsson, 2013*; *Vickerton, Jarvis & Jeffery, 2013*). Both high concentration of iodine solutions (*Vickerton, Jarvis & Jeffery, 2013*) and greater durations of immersion for staining (*Gignac et al., 2016*) may cause more extreme shrinkage. Based on our experience, we currently prefer to use lower concentrations of Lugol's solution (usually 1%) for specimens of this size, and we perform test scans (as possible) to ensure adequate staining and avoid extended, unnecessary immersion which may result in further shrinkage of the tissue. In addition, some authors have observed differential shrinkage among different tissue types, such as the brain and eyes (*Vickerton, Jarvis & Jeffery, 2013*; *Hedrick et al., 2018*). Staining isolated tissues samples also causes more extreme shrinkage (*Vickerton, Jarvis & Jeffery, 2013*), while iodine staining of whole specimens is known to produce far less dramatic reductions (*Tahara & Larsson, 2013*; *Hedrick et al., 2018*). Our results confirm that diceCT is associated with reductions in epithelial perimeters as well, by approximately 3% to 5%. This is similar to a 5% reduction in embryonic quail cranial length following iodine staining (*Tahara & Larsson, 2013*).

The 11–14 % differences between μ-CT-based and histology-based perimeters most likely reflects additional shrinkage of the tissue during graded ethanol baths prior to paraffin embedding (*Tahara & Larsson, 2013*). This large artefactual distortion means previous quantitative studies of epithelia, at least those based on paraffin-embedding of fixed decalcified tissues (*e.g.*, *Adams, 1972*; *Bhatnagar & Kallen, 1975*; *Gross et al., 1982*; *Smith & Rossie, 2008*), likely report distortions of epithelial surface areas of nasal fossa structures. These may be underestimations for external perimeters (*e.g.*, the epiturbinal and maxilloturbinal described here), or overestimations for internal perimeters (*e.g.*, the roof/septum described here). It may be notable that turbinals are supported by especially thin bone, and such structures may shrink to a greater extent than other

surfaces with more substantial support, such as the peripheral contours of the nasal fossa (*e.g.*, septum). Indeed, in one recent study we corrected for shrinkage of the rostral projection of the first ethmoturbinal (*DeLeon & Smith, 2014*).

While no method other than scans of fresh tissues can be expected to eliminate shrinkage, both diceCT and histology provide a powerful means of tissue differentiation. If epithelial measurements using diceCT can match or approach the accuracy of histology for epithelial tissue identification, then it would have a great advantage of far less shrinkage artifacts when used for the study of whole specimens. Thus, our findings demonstrate the great potential of diceCT for studying rare, valuable specimens (*e.g.*, museum samples) nondestructively, and with less distortions than is seen using histology (*DeLeon & Smith, 2014*). In addition, we demonstrate that iodine staining followed by stain removal using sodium thiosulfate does not interfere with histological study of well-preserved specimens using traditional techniques such as trichrome staining.

## Identification of epithelia using histology and diceCT

In both bat species, OE is easily identifiable based on well-established light microscopic characteristics such as the presence of rows of cell bodies of olfactory sensory neurons and elongated cilia (*e.g.*, *Chamanza & Wright, 2015*; *Dennis et al., 2015*). The latter are not individually observable by light microscopy, but do stain as a narrow band whereas tangled cilia exist within a mucous film. In contrast, the shorter kinocilia of respiratory epithelium are identifiable at higher magnification.

Another feature of OE is its greater relative thickness, on average, compared to most non-olfactory types (*Dieulafé, 1906*). This relates to the distinctive lamina propria that supports it, in which Bowman's glands and bundles of olfactory axons are nested (*Chamanza & Wright, 2015*; *Dennis et al., 2015*). However, in the adult *Cynopterus* there is a wide range of OE thickness (26.3 to 71.6 μm) and significant differences among sampled structures in the mean OE thickness (Table 2). This result agrees with findings on large samples of postnatal rats, in which OE has a similarly wide range in thickness (*Weiler & Farbman, 1997*). *Weiler & Farbman (1997)* also found regional variation in thickness, noting that OE on convex structures was typically thicker than that on concave structures. However, note that here we observed thinner OE along the convex peripheral edges of turbinals.

The regional variation in OE thickness, which could be typical of mammals, complicates our ability to use epithelial thickness as a criterion for annotating OE limits. A bigger limitation is that the transition of OE to ciliated respiratory epithelium can be difficult to detect (*Yohe, Hoffmann & Curtis, 2018*), as was the case in our study. As epithelia become thinner, they may closely approach voxel size. This means transitions must be abrupt to be accurately detected. Nonetheless, OE may be clearly detectable based on relative thickness and its greater degree of radioopacity compared to adjacent respiratory epithelium, even if its precise boundaries are not detectable. Our observations, supported by statistical results, indicate OE could be reliably identified blindly in the adult *Cynopterus*, but not in the fetal *Desmodus*. The inability to identify OE in *Desmodus* was likely related to the small size of the specimen and perhaps less differentiation of the OE.

## Identification of mucosae using histology and diceCT

Compared to the sole use of epithelial thickness in annotation of OE on diceCT images, adding the criterion of lamina propria radioopacity yields a better match of perimeters to that of histology-informed perimeter measures. Although paired perimeter measurements suggest thickness alone was a highly effective criterion for identifying OE in the adult bat, blind annotations included some unusually thick respiratory epithelium, overestimating the amount of OE rostrally (Fig. S3). The combined criteria for blind annotations produced a better match rostrally (Fig. 4). OE thickness produced a very poor match in perimeter measurements in the fetal bat, while the combined criteria led to identification of OE in precisely the same range of slices as the histology-annotations, with a very close correspondence of perimeter measures (Fig. 4). The criteria used to assess the entire mucosa (epithelium plus lamina propria) may be suitable for analysis mucosae of immature individuals comprising a cross-sectional age sample. This will need to be assessed using earlier stages of prenatal animals.

Our study suggests glandular tissue adds to radioopacity after iodine infiltration, as does covering epithelium. This was noted also by *Yohe, Hoffmann & Curtis (2018)*. In this respect, it should be noted that thickness of olfactory and respiratory mucosae varies greatly and can overlap in range of thickness. *Smith et al. (2021)* related this mostly to the composition of the lamina propria. Respiratory mucosa has thickened lamina propria when it is highly vascular or highly glandular. In the former case, large venous sinuses may be visible (as is seen in the adult bat studied here (Fig. 2C). In either instance, the epithelia of such mucosae are often thin; and because these epithelia are closer to voxel dimensions they may be poorly resolved. On the other hand, olfactory mucosa has a broad range of thickness with regional variation (*Smith et al., 2021*). Bowman's glands are a reliable indicator, but the amount of glandular tissue may vary; this can relate to differences in mucosa thickness, as seen in the convex *versus* concave sides of some turbinals (*e.g.*, see Fig. 3D). This tissue-level complexity means that an observer may be forced to occasionally rely on epithelium thickness alone as a criterion for blind identification of OE. However, the availability of a representative histological specimen is essential for interpretations.

Certain limitations of the present study will require additional scrutiny. The better visualized epithelia in diceCT of *Cynopterus* compared to *Desmodus* seems quite explainable based on the thinner OE in the latter (very near voxel size). However, Bowman's glands were not as discretely visible in our study compared to respiratory glands identified by *Yohe, Hoffmann & Curtis (2018)* using diceCT (see Fig. 2, therein). It is notable that *Yohe, Hoffmann & Curtis (2018)* used about twice the concentration of iodine and longer durations of staining compared to the present study. Therefore, future studies should explore different durations of iodine staining for effectiveness in identification of epithelia and Bowman's glands. On the other hand, *Yohe, Hoffmann & Curtis (2018)* used specimens stored in 10% formalin, while here we examined specimens stored for decades in ethanol. So another area of exploration should be the effectiveness of diceCT for identification or nasal glands and epithelia in specimens stored in different fixative (see further discussion in *Hedrick et al., 2018*).

## CONCLUSIONS

Although diceCT is, as yet, only a match for light microscopy at low magnifications, our study indicates diceCT slices offer a valuable tool to annotate transitions in mucosa type within the nasal cavity. Reliance on epithelial thickness alone may suffice as an identifier of OE, particularly in the case of specimens that are well-stained, with mature, relatively thick olfactory epithelium, and given sufficient resolution. However, the use of combined criteria that interpret glandular composition of the lamina propria, along with epithelial thickness, helps to avoid false positive identification. In addition, immature specimens may exhibit characteristics of olfactory glands that can aid in identification of olfactory mucosa, even when the olfactory epithelium by itself is not completely discrete, as shown here with a fetal bat. We suggest that histology from one reference specimen of the species would be sufficient to aid in detecting epithelial transitions using diceCT.

Thus, diceCT can greatly reduce destructive methods, and at the same time greatly increase sample sizes, with less artefactual changes than occurs with histological processing. A combination of diceCT and μCT of the same specimens will allow a fuller understanding of what type(s) of mucosa line each turbinal. This would provide a firmer basis, or cautionary caveats, for the use of individual bones such as turbinals as proxies for a particular function (*e.g.*, *Van Valkenburgh et al., 2014*; *Martinez et al., 2018*). This also has important application to future quantitative studies to further our understanding of the link between OE surface area and ecological variables (*e.g.*, *Yee et al., 2016*), and in the study of fluid dynamics in the nasal airways (*Craven, Paterson & Settles, 2010*; *Ranslow et al., 2014*).

## ACKNOWLEDGEMENTS

We are grateful to Adam Hartstone-Rose, Deborah Bird, and one anonymous reviewer for numerous constructive comments that substantially improved our manuscript. We also thank Chris Vinyard for scanning one of the specimens used in this study.

### Funding

This work was funded by a grant from the College of Health, Environment, and Science, Slippery Rock University and by NSF grant # BCS-1830894 and BCS-1830919. There was no additional external funding received for this study. The funders had no role in study design, data collection and analysis, decision to publish, or preparation of the manuscript.

### Grant Disclosures

The following grant information was disclosed by the authors:
College of Health, Environment, and Science, Slippery Rock University: BCS-1830894 and BCS-1830919.

## Competing Interests

The authors declare that they have no competing interests.

## Author Contributions

- Timothy D. Smith conceived and designed the experiments, performed the experiments, analyzed the data, prepared figures and/or tables, authored or reviewed drafts of the paper, and approved the final draft.
- Hayley M. Corbin performed the experiments, authored or reviewed drafts of the paper, helped to write grant which funded research, and approved the final draft.
- Scot E. E. King performed the experiments, authored or reviewed drafts of the paper, and approved the final draft.
- Kunwar P. Bhatnagar conceived and designed the experiments, authored or reviewed drafts of the paper, originally obtained specimens used in the study, and approved the final draft.
- Valerie B. DeLeon conceived and designed the experiments, performed the experiments, authored or reviewed drafts of the paper, and approved the final draft.

## Animal Ethics

The following information was supplied relating to ethical approvals (*i.e.*, approving body and any reference numbers):

The Institutional Animal Care and Use Committee at Slippery Rock University approved this research (Protocol #: 2021-03T).

## Data Availability

Tabulated measurements are available in the Supplemental Files.

All CT scan slices are available on MorphoSource:

https://doi.org/10.17602/M2/M365396

https://doi.org/10.17602/M2/M368062

https://doi.org/10.17602/M2/M365511

https://doi.org/10.17602/M2/M365442

https://doi.org/10.17602/M2/M365381

https://doi.org/10.17602/M2/M389419

https://doi.org/10.17602/M2/M367225

https://doi.org/10.17602/M2/M367209.

## Supplemental Information

Supplemental information for this article can be found online at http://dx.doi.org/10.7717/peerj.12261#supplemental-information.

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
