# Peer review of "A comparison of diceCT and histology for determination of nasal epithelial type"

_PeerJ, doi:10.7717/peerj.12261_

## Round 0.1 · original submission · Major Revisions

The authors need to pay careful attention to all reviewers' comments. The authors should consider these comments and corrections in submitting a revised manuscript.

·

Basic reporting

This paper is clearly written, well referenced, professional, and altogether well reported.

Experimental design

This paper has excellent experimental design.

Validity of the findings

I have no doubt about the validity of these findings.

Additional comments

In this fantastic paper, Smith and colleagues study the olfactory epithelium of two bats comparing diceCT and histological methods. This paper is excellent, and should be published largely as it is. Frankly, I find it hard to find substantial fault in any aspect of this paper, but, in an effort to provide SOME constructive feedback, there are a couple of notes that might be worth mentioning. By no means should any of these modifications be required, and, in my opinion, the paper should be accepted essentially in its current form with one notable caveat: as the authors know, I know absolutely nothing about epithelial tissues! If a colleague with that expertise finds faults, those could/should be taken seriously.

With that said, here are some thoughts:

1) There are some inconsistencies in tabs and spaces throughout the paper. In our era of copyediting, this does not really bear mentioning, but I am grasping at straws to find flaws here!

2) It might be worth highlighting that the investigators chose to increase the concentration of iodine rather than the duration of staining of their larger specimen for their diceCT methods. To my knowledge, the jury is still out for which of these yields better results (I’m not aware of whether this sorely needed comparative study has been accomplished, but I would love to collaborate with any of the authors on such an investigation!), but in our anecdotal experience, we believe that increasing duration instead of concentration causes less shrinkage. Since shrinkage was explicitly of interest in this study, it is probably worth mentioning this as a caveat, though, since there really is no scientifically definitive answer, this is certainly not a flaw of the current investigation. If the authors are aware of a reason to choose concentration over duration, then it would be valuable for the field if they would mention that!

3) On a related note, the investigators should probably mention the potential shrinking effects that the specimens have undergone during their decades of storage. The two recent papers by Leonard and colleagues (though about muscles) speak to this. Of course, we have no idea what the cumulative effect of decades of preservation, then diceCT, then slide mounting has on these tissue dimensions, and this is certainly not a disqualifying aspect of this paper, but it is probably worth mentioning as a caveat.

4) As the investigators note, it is problematic that some of their tissues of interest approach their voxel sizes. Indeed this is a problem all of us face. It might be worth mentioning that surely technology will solve this problem: CT resolution is getting better and better as are staining regimes and hopefully too digital dissection software. This problem is important but temporary.

5) The authors note that histology has important uses that are beyond what we can do with CT, but they don’t note that, theoretically, serial histology slices could be used by those of us who do volumetric based “digital dissection” too. This was important in the early 3D anatomy programs (e.g., I think that the first digital cadavers were made using serial slices that could then be resliced in any plane?), and may still be the way that digital cadavers are assembled. I haven’t seen us anatomists do this (e.g., making stacks from slides and resampling their planes in 3D space), but we could, right? And maybe we should?

6) Arguably the only thing that, in my opinion, MUST be added to this paper is a better highlight of some of the truly important findings of this paper that the authors never mention: for instance, to my knowledge, this is the best (if not first?) study to demonstrate a) how well diceCT can be used on specimens that have been stored for DECADES in fluids (though I would appreciate more details about how these specimens were initially preserved and stored) and b) that diceCT specimens can subsequently also be histologically stained. Both of these are seriously important pieces of information! DiceCT is resource intensive both in terms of materials and time and I have been reluctant to try it on specimens that were not optimal. Specifically, I was nervous that the long-term storage might have made the tissues less permeable to the iodine. And I am astonished that the iodine and its subsequent de-stain did not undermine the ability to histologically stain those tissues. Maybe other people have found both of these previously, though I am unaware of those findings (I do feel like I read about old museum specimens in diceCT, but I can’t remember where), and it is certainly worth highlighting that this paper shows that these methods do indeed work in these specimens and in combination.

Again, this paper is fantastic, and I look forward to seeing it published. – Adam Hartstone-Rose

·

Basic reporting

1. Clear and unambiguous, professional English used throughout
The manuscript is written in correct English. General suggestion I have is that the goals and methods need to be edited by the authors to focus the language to better orient the readers to the anatomy, describe the particulars of each test type and make the relevance of the paper clearer. More details on this are below.

2. Literature references, sufficient field background/context provided
No comment

3. Professional article structure, figures, tables. Raw data shared
The article follows an acceptable format.

The figures serve the text in general. More needs to be done to orient the reader to the anatomical context of the figures, and the figures need to be labeled and annotated better to help clarify the tests performed by the authors. For example, I think the annotations the authors made on the mucosa types need to be made more prominent and evident. Fig S1 shows the sights of sampling, and Fig.S2 shows some boundaries of their annotations, but none of this is present in the main figures. I believe it needs to be. All figures are insufficiently labeled. Here are a number of suggestions regarding the figures:
Figure 1:
---Please clarify for the reader where in the skull we are in these sections.
---What does v in vETII mean?
---Do ETI and ETII correspond to anatomical terms? I think then ETII would be ventral to ETI? If they don’t correspond, make that clear that these are labels for this study alone.
---Labels and lines are displaced in a number of places. Epiturbinal is outside the nasal cavity, for example.
---What is ET vs ETI or ETII or vETII?
---Define NpD in the legend.
---Note the white lines with no labels.
---Is that the cranial cavity? If so, please label it for orientation.
Figure 2:
---On this or another introductory main figure, label parts of mucosa as background for readers and because the various features are central to the different tests the authors use: Epithelium, cilia, borders of lamina propria for both olfactory and respiratory mucosa, etc.
Figure 3:
--- Radiopacity is mentioned in Fig. 3. In the text, radiopacity is also referred to in terms of glandular tissue. If possible, please feature and label this in this figure or another in order to illustrate for the reader what kind of assay the authors use for their blind annotations.
---The legend says that the ‘turbinal is almost completely radiopaque’. This is confusing because “turbinal” is bone. Using turbinal interchangeably as just the bone and the entire structure with accompanying soft tissue can be confusing.
---Some labels seem to be displaced, e.g. GL
---Bowman’s gland is BG and GL on various figures.
Figure 4:
---Perhaps this figure could include a comparison between blind test #1 and #2 to show the difference in the results when the added criterion of radiopacity is added. This would help focus the conclusions of this article: the success of using diceCT in lieu of histology and the limitations of basing estimates from diceCT on epithelial thickness alone.
Figure S1:
---Which specimen is this?
---It’s not clear to me if c and d are meant to be olfactory mucosa or were chosen as respiratory mucosa locations. Please clarify in the legend and text.
Figure S2:
---Perhaps enlarge S2b to include all four annotations and omit c.
---Again, make it clearer, by coloring or something what is meant by perimeter.
---Label lamina propria on diceCT image.
---Where is S2i from in S2h? Please label S2i with features that distinguish it from olfactory tissue in S2g. They look very similar. For example, what distinguishes respiratory kinocilia, which is mentioned as an indicator in the legend, from OE cilia?
Figure S3:
---Please make it clear which trial this is.

4. Self-contained with relevant results to hypotheses
I think this study stands on its own and is an appropriate unit of publication.

Experimental design

1. Original primary research within Aims and Scope of the journal.
I believe this article does meet the aims and scope of the journal.

2. Research question well defined, relevant & meaningful. It is stated how research fills an identified knowledge gap.
I think the authors accomplished their research goal, and it is relevant and meaningful. I also think the authors could edit the Introduction and Abstract to better focus the readers’ attention to the fact that previous studies using diceCT did not test the reliability of diceCT-generated annotations in relationship to histological ground truth by performing blind annotations, leaving an important gap that they begin to fill with this article.
The end of the Conclusion reminds us that using diceCT to identify and estimate mucosal perimeters would quicken the work of researchers like Yee, 2016 and Craven, 2010, etc in quantitative studies of the link between OE surface and ecological variables and airflow models and olfactory function. These examples and other publications on anatomical correlates of olfactory function give particular relevance to this current study, and perhaps these might be referenced in the Introduction as well to better illustrate the necessity of this study.

3. Rigorous investigation performed to a high technical & ethical standard.
I think this investigation was performed with rigor and to a high technical and ethical standard.

4. Methods described with sufficient detail & information to replicate.
The methods may need to be focused and pared down to define each test and clarify the protocol steps. The use of radiopacity in estimating tissue perimeters needs to be clarified so as to be replicable. Authors might also clarify how indicators of glandular tissue are identified and used in the blind annotations.

Validity of the findings

1. Impact and novelty not assessed. Meaningful replication encouraged where rationale & benefit to literature is clearly stated.
Although this article contributes new anatomical data on the nasal cavities of two bat species, it seems to be first and foremost a methods paper. Because its small sample is small and criteria are sometimes qualitative, it an initial study that establishes an good first step toward a working protocol for discerning and quantitatively estimating relative surface areas of mucosal types in mammal noses from diceCT images.

2. All underlying data have been provided; they are robust, statistically sound, & controlled.
I suggest the authors speak to their sample of two individuals, why it was necessary and how a goal of future work might be to increase the number of specimens from a single species in order to statistically verify their results.

3. Conclusions are well stated, linked to original research question & limited to supporting results.
The Results section needs clarification in a number of places. First, the subsections should more clearly track the series of tests/trials set up in Materials and Methods and reinforce each question at the beginning of each subsection. This will facilitate clarity and narrative style. Figures need to be referenced more often in the Results section to to reinforce the results visually. Referencing tables is not enough. The Discussion section is more narrative and flows better.

--- Paragraph starting Line 212: start paragraph with a sentence to reiterates question for the reader; Clarify annotation type here; define NT, vestibule, free margins; clarify what “five locations” correspond to; refer to figures; clarify when the text is referring to OE and when to non-OE.

----Paragraph starting Line 231: Again, reinforce question and trial type for reader in first sentence; artefactual; clarify what’s been measured here; name “two structures”; is this paragraph repeated by the next?

---Line 264: Subheading could be more informative
---Line 284: Clarify for the reader how to discern on Fig.3c that the lamina propria is highly glandular. I don’t see it clearly.
---Line 285: Label small patches of non-OE in Fig. 3. I don’t recognize them.
---Line 289: The subheading is awkward. I’m confused by Trial 1. Did the previous results have anything to do with Trial 1? If not, it would be helpful to make it clear that all Results up to this point were histologically-informed and that this is the first departure from that.
---Line 292: The figures references didn’t seem to match the text.
---Line 295: Avoid vague descriptions like ‘both structures.’
---Line 302: The summary of the repeated measures ANOVA was not clear to me. Also, which types does “annotation type” include here?
---Line 307: I couldn’t find Table S3, S4 or S5.
---Line 320: Subheading doesn’t seem to summarize Trial 2.
---Line 321: ‘Entire mucosa’ could mean various things.
---Line 325: The second type of blind annotation is referred to here as ‘combined criteria of epithelial and overall mucosal thickness.’ This doesn’t describe the criteria well.
---Line 330: Name ‘both regions.’
---Line 368: Mucosa and epithelium are sometimes used interchangeably and sometimes as distinguishing identifiers. Lines 368-371 settle that puzzle, so this needs to appear sooner in the text.
---Line 395: Name the ‘two structures.’
---Line 417: The use of the word turbinals needs clarifying throughout. For example, here, “turbinals are supported by especially thin bone, and such structures may shrink to a greater extent than other surfaces with more substantial support.” the turbinal seems to be referred to as a whole unit supported by bone, whereas I understand a turbinal to be the skeleton itself. There are other examples of this.

4. Speculation is welcome, but should be identified as such.
I see no speculation in this article.

Additional comments

Just by way of summarizing what I took away from the article:

In “A comparison of diceCT and histology for determination of nasal epithelial type” Smith et al. test the potential for diffusible iodine-based contrast-enhanced computed tomography (diceCT) to replace traditional methods, such as microCT and histology with a non-destructive method for identifying olfactory mucosa in mammalian nasal cavities. The authors chose to study two specimens of bats, one an adult fruit bat and the other a full-term fetal vampire bat. They designed two methods of blind annotation of olfactory mucosa on the diceCT images, one based mucosal thickness alone (an indicator known in the literature) and one based on mucosal plus relative radiopacity (brightness here) and the putative presence of a dense presence of radiopaque glandular cells in the lamina propria underlying the epithelium. This was done at various constant anatomical landmarks on the ethmoturbinals and nasal septum throughout the image series. The authors successfully aligned three image types (histological sections, microCT and diceCT) and used a histologically-informed estimate of olfactory mucosa against which to judge their blind estimates. They concluded that in the posterior extent of the nasal cavity, that is, caudal to the most anterior extent of the ethmoturbinals, the blind estimates of olfactory mucosa perimeter from diceCT images based on thickness alone were not significantly different from estimates from histological samples. Rostral to this, the relationship fell apart due to highly vascularized, thick respiratory epithelium. This shows the limits of this one method of annotation. When radiopacity was taken into consideration, blind estimates of olfactory mucosa from diceCT closely matched those done from the histological sections throughout the nasal cavity. The authors also confirmed and quantified tissue shrinkage resulting from histological and iodine-stained specimens.

This study successfully defines the potential and some of the limits of using diceCT to discern and quantify olfactory and respiratory mucosa and will likely quicken the work of researchers interested in the mechanics and evolution of respiration and olfaction by guiding them in the use of diceCT, particularly with rare specimens that need to be investigated non-destructively.

My main suggestion is that the authors consider giving the reader a firmer footing in each question posed in the article and the particulars/criteria of each attending test. The language can be pared to focus on just that. I spent some time while reading the article having to double check where I was in the skull, the mucosa, and in the sequence of goals/tests. Please clarify this, label every term referenced, reinforce the goal and particular criteria of each step, It needs to be easier to follow.

Additionally, please acknowledge the small sample and the qualitative nature of a criterion such as radiopacity and reassure the reader that this is the first step toward working out a replicable protocol for this important work.

Reviewer 3 ·

Basic reporting

No comment

Experimental design

No comment

Validity of the findings

No comment

Additional comments

Smith, et al. investigate methodological differences of inferring olfactory versus non-olfactory epithelia in the nasal cavity of bats using histology, µCT, and diceCT. As diceCT methods launch the field of comparative anatomy into directions never before possible, it is critical to note the assumptions and effects of diceCT on the detailed anatomy that can is observed from histology. Despite diceCT going “mainstream” in 2016, very few studies have actually properly compared histology and diceCT within the same specimen, let alone for olfactory epithelia. Thus, I commend the authors for their efforts in doing so and advocate that this study be published if authors can address a few of my concerns below.

Major comments:
First, and perhaps this is a major-minor comment, there can be some broader implications discussed or introduced. For example, many studies estimating olfactory epithelia use turbinate reconstructions directly from bone (Van Valkenburgh, et al. 2011, Journal of Anatomy; Martinez, et al. 2020, PNAS). These studies often acknowledge that they are making a large assumption that there is direct correlation between the olfactory tissue and the turbinates, but this is rarely quantitatively verified. I feel like this study offers that opportunity to more clearly articulate this. Establishing these boundaries is essential for the use of OE studies in specimens where soft tissue specimens are not available. I suggest integrating this more into the narrative of the study.

My second concern regards length of staining time. Seven days for both specimens (albeit at different concentrations) concerns me that increased visibility of neurons and glands within the non-olfactory epithelia are minimal relative to Yohe, Hoffmann, and Curtis (2018), which stained in double the concentration for up to 15 weeks. Also, I do imagine that the thickness and density of the Cynopterus is significantly greater than that of the embryonic Desmodus, which again would affect the rate of diffusion. While I do not think this affects the ability to detect the very bright olfactory epithelium, I do think there is much commentary on the presence of these mucosal glands as guiding the researcher for non-OE. Yet, from the figures I do not see any good examples of this and I wonder if this is due to undersaturation of iodine in the specimens. While I do not expect the researchers to repeat the study, I think a bit of discussion is warranted in the context of comparing to other diceCT studies using bat specimens (i.e., Hedrick, et al. 2018, Hall, et al. 2021 Evolution). I also recommend in the future for the researchers to explore longer stain times or to continuously refresh the iodine solution until the solution no longer discolors (suggesting more complete diffusion).

Third, unless I missed it, the authors do not comment on the effect of the diceCT stain on the histology. I know that it was unstained in sodium thiosulfate, but did this really remove all of the iodine? Could any be detected in the histological sections? Several of the authors have many publications using H&E on bats...how did this staining compare to previous (e.g., in coloration)? The reason I bring this up (outside of genuine curiosity), is that often times people discuss how they think the iodine will affect the histology with the assumption it would negatively do so, and then fail to ever do the histology because of this unwarranted assumption. I would like to see the authors comment on this in the discussion, as it would minimize poorly established rumors as to whether or not acceptable histological staining and sectioning is possible after diceCT.


Minor comments:
L61: I think a couple of sentences introducing the types of epithelia in the mammalian nasal cavity is warranted, along with what types of cells, general anatomical features, etc. are associated with these different types.

L159: “dice CT” should be one word

L302: “nut” should be “but”

L546: Typo in authors name...”Davie” should be “Davies”

Scale bars on the supplement Fig 1 would be appreciated

---

## Round 0.2 · accepted · Accept

The authors have paid careful attention to the reviewers' comments. The manuscript has been substantially revised in accordance with the comments of the reviewers.

·

Basic reporting

No comment

Experimental design

No comment

Validity of the findings

No comment

Additional comments

I appreciate how meticulously the authors considered the reviewers' comments and made multiple positive changes to the paper. I believe the authors took an already very good paper and made it even stronger. This will be a wonderful contribution.

Reviewer 3 ·

Basic reporting

no comment

Experimental design

no comment

Validity of the findings

no comment

Additional comments

The authors did a great job at responding to both my concerns and the concerns of other reviewers. I have no further comments and look forward to the manuscript being published.